# Unravelling the Initial Triggers of *Botrytis cinerea* Infection: First Description of Its Surfactome

**DOI:** 10.3390/jof7121021

**Published:** 2021-11-29

**Authors:** Almudena Escobar-Niño, Rafael Carrasco-Reinado, Inés M. Morano, Jesús M. Cantoral, Francisco J. Fernandez-Acero

**Affiliations:** Microbiology Laboratory, Institute for Viticulture and Agri-Food Research (IVAGRO), Marine and Environmental Science Faculty, University of Cádiz, 11510 Puerto Real, Spain; almudena.escobar@uca.es (A.E.-N.); rafael.carrasco@uca.es (R.C.-R.); inesm.morano@gmail.com (I.M.M.); jesusmanuel.cantoral@uca.es (J.M.C.)

**Keywords:** *Botrytis cinerea*, fungal phytopathogen, surfactome, proteomics, signaling cascades

## Abstract

*Botrytis cinerea* is a critically important phytopathogenic fungus, causing devastating crop losses; signal transduction cascades mediate the “dialogue” among the fungus, plant, and environment. Surface proteins play important roles as front-line receptors. We report the first description of the surfactome of a filamentous fungus. To obtain a complete view of these cascades during infection of *B. cinerea*, its surfactome has been described by optimization of the “shaving” process and LC–MS/MS at two different infection stages, and with both rapid and late responses to environmental changes. The best results were obtained using PBS buffer in the “shaving” protocol. The surfactome obtained comprises 1010 identified proteins. These have been categorized by gene ontology and protein–protein interactions to reveal new potential pathogenicity/virulence factors. From these data, the percentage of total proteins predicted for the genome of the fungus represented by proteins identified in this and other proteomics studies is calculated at 54%, a big increase over the previous 12%. The new data may be crucial for understanding better its biological activity and pathogenicity. Given its extensive exposure to plants and environmental conditions, the surfactome presents innumerable opportunities for interactions between the fungus and external elements, which should offer the best targets for fungicide development.

## 1. Introduction

The life cycle of fungal plant pathogens consists of invading plant tissues and transforming vegetable biomass so that the fungus can live, spread, and reproduce its active structures. In effect, a molecular competition takes place between plant defense strategies and fungal virulence/pathogenicity factors. The specific environmental conditions in which this competition occurs play a key role during the process. Of all the fungal phytopathogens, *Botrytis cinerea* has been considered the most harmful in the western hemisphere [1], not only for the damage caused to hundreds of crops of agronomic importance, but also because it serves as a molecular model, and there is increasing commercial interest by industry in the development of new environmentally-friendly fungicides [2].

*B. cinerea* owes its pathogenic potential to a large group of genes known as pathogenicity or virulence factors. Those genes have been listed in the plant–host interaction database [3]. As a pathogenic model, *B. cinerea* has been used to develop defective mutants in those factors in order to elucidate the role of each during the infection process. However, only 148 genes have been genetically studied by mutant analysis [4], representing only 1.3% of the total number of proteins predicted from the genome of *B. cinerea* [5]. Mutant analysis is therefore a serious bottleneck in protein research, and several other approaches have been developed based on “…omics” methodologies. From all of the available alternatives, proteomics approaches are now considered the most promising methodology given their potential for finding new pathogenicity/virulence factors without the need to determine a priori candidates. Against other “omics” methodologies, proteomics is proposed to be the most relevant level of analysis [6].

Several proteomics studies have been conducted to unravel the infective mechanisms of *B. cinerea*, most of them based on synthetic culture media [7]. In order to clarify the changes in the phenotype of *B. cinerea* when plant-based elicitors were used as a sole carbon source, those synthetic media were replaced by glucose as a constitutive stage, and deproteinized tomato cell walls (dTCW) as a virulence inductor [8]. The results showed crucial changes in the phenotypes observed, suggesting a connection between the gene expression of the virulence factors and culture conditions [9]. While the suggested constitutive stage using glucose seems to inhibit the production of cell-wall-degrading enzymes (CWDE), preventing the production of several virulence factors, the production of *B. cinerea* toxins seems to be induced, with a maximum production of botrydial and dihydrobotrydial after 5 days [10]. On the other hand, when dTCW is used, a number of CWDEs are triggered but the synthesis of toxins, including the expression of the gene BcBOT2, is halted [9]. This finding shows the very close relation between the pathogen and the environmental conditions: the fungus modulates its infective mechanisms to specific situations.

The connections among plant, environmental conditions, and fungal invasion activities are regulated by signal transduction cascades. Surface proteins transmit the signal from the exterior across membranes to the cytosol and nucleus, where cellular response is initiated [11]. From external signals (primary messengers), the information flow continues to involve a large number of components such as receptors, protein phosphorylation processes, secondary messengers, transcription factors, and more. Using classical molecular approaches, these cascades have been widely studied and described in *B. cinerea*; examples include monomeric G proteins, G protein, cAMP, calcium mediation and MAP kinases. With respect to phytology, these signaling cascades are relevant because most of their known components have been classified as virulence/pathogenicity factors, thus highlighting their crucial role during the infection process.

With the objective of increasing our knowledge about the protein components of *B. cinerea* signaling cascades, various proteomics studies have been carried out. Using different plant-based elicitors, phosphoproteome analysis of *B. cinerea* has been described as one of the main processes of information transference [12,13]; specific phosphorylation patterns are presented depending on the carbon source used. The *B. cinerea* membranome has also been described; changes in its composition with the increase of signaling function in glucose conditions and in the carbohydrate degradation process in TCW conditions have been revealed [14]. Overlapping both pictures, a new study has been made by purifying those membrane proteins controlled by phosphorylation. From the 1112 membrane-associated phosphoproteins previously identified, those differences have been associated with various processes, including pyruvate metabolism, unfolded protein response, oxidative stress response, autophagy and cell death [15]. In silico analysis of these processes has revealed the relevance of TOR signaling, the phosphor-relay signal transduction system, and inositol lipid-mediated signaling, specifically in GLU conditions. In contrast, however, calcium-mediated signaling GO annotation is only present among the proteins identified in the TCW condition [16].

However, of the predicted total number of proteins in the genome of *B cinerea*, the percentage identified in proteomics studies is only around 12% [15]; new calculations with the latest genome annotations and the development of new proteomic approaches to identify many more proteins, especially those components involved in signaling cascades, are necessary. This is the object of the work reported here.

The primary contact between *B. cinerea* and its host takes place at the cell surface [17] where the initial steps of signaling, adhesion, transport, etc., are initiated. In pathogenic microorganisms, proteins located at the cell surface are of special interest; this is where the processes of host recognition, and initial steps of invasion, toxin production, etc., must be triggered. This relatively large group of proteins can be considered a subproteome; these surface-associated proteins have been described as among the most variable and complex [18], and are especially difficult to work with [19] compared with other types of protein, mainly due to their low abundance and insolubility. Specific protocols have been developed to unravel this surfactome avoiding cell lysis; most of the protocols are based on the controlled use of proteases that digest the external domains of the proteins, coupled with an LC–MS/MS platform for peptide identification. To date, the surfactomes of only 10 g positive bacteria, 8 g negative bacteria, and 6 eukaryotes, have been described [19], and none of them is a filamentous fungi. It is hoped that the study of these proteins may help to unravel the biological processes taking place during the initial infection stages, and that new proteins can be identified as potential therapeutic targets

The main aim of the present work is to optimize the protocol to obtain the surfactome of the phytopathogenic fungi *B. cinerea*, and identify the proteins exposed on the cell surface during a constitutive stage, using glucose as a sole carbon source, and during induction of pathogenicity using deproteinized tomato cell walls, both as a rapid response with incubation for 2 h (2 hpi), and as a late response with incubation for 48 h (48 hpi).

## 2. Materials and Methods

### 2.1. Fungal Strains and Culture Conditions

*B. cinerea* B05.10 was provided by the Spanish Type Culture Collection (CECT). Conidial stock suspensions were prepared and maintained as previously reported [8]. Two different carbon sources were used: glucose (GLU) (Panreac, Barcelona, Spain) as the constitutive stage; and deproteinized tomato cell walls (TCW) as the virulence inductor, as previously described [8]. In brief, deproteinized tomato cell walls (TCW) are prepared by grinding lyophilized tomatoes to a fine powder in liquid nitrogen with the aid of a mortar and pestle. The frozen powder was then washed for 5 min with approximately 2.5 volumes of cold 0.1 M of potassium phosphate buffer (pH 7), insoluble material is recovered by filtration, and the process is repeated five times. This step repeated with NaCl (1M), distilled water, chloroform-methanol (1:1; five times), cold acetone (three times), and ethyl acetate. The residue remaining, which constitutes the cell walls, was air dried and stored at 4 °C.

Five hundred milliliter flasks containing 250 mL of minimal salt medium (MSM) (50 mM NH_4_Cl, 7.3 mM KH_2_PO_4_, 4.2 mM MgSO_4_, 6.7 mM KCl, 0.07 mM FeSO_4_) supplemented with 1% of GLU were inoculated with *B. cinerea* conidia, to a final concentration of 5 × 10^4^ conidia/mL. After 3 days, fungal mycelium was harvested by filtration in a 30-µm nylon filter (Sefar Nytal, Heiden, Switzerland) and transferred to new 500 mL flasks containing 250 mL of MSM supplemented with 1% of deproteinized TCW. After two hours, the mycelia were harvested by filtration for rapid response. For late response, the mycelia were harvested after 48 h. PhosSTOP Phosphatase Inhibitor Cocktail (Roche, Basel, Switzerland) was added to the culture according to the manufacturer’s instructions. Three independent replicas were assayed for each culture condition. Replicates were incubated in parallel at 180 rpm at 22 °C under alternating 12-h light/dark cycles.

### 2.2. Optimization of “Shaving” Process in B. cinerea

For the optimization of surface protein extraction (“shaving”), two different protocols were compared: first, the protocol based on phosphate-buffer saline (PBS) plus sucrose for the study of *Streptococcus pneumonia* [20] and, second, the protocol based on ammonium bicarbonate buffer previously used for *Candida parapsilosis* and *Candida tropicalis* [18]. These protocols, using two different buffers, were modified to obtain the first analysis of the surface receptors of *B. cinerea* by shaving. For the shaving optimization procedure, Erlenmeyer flasks of 500 mL with 250 mL of PDB medium (Potato Dextrose Broth; Scharlau, Barcelona, Spain) inoculated with 5104 conidia/mL, were used. Three biological replicas were incubated for 5 days, with a photoperiod of 12 h, at 22 °C and 180 rpm (Figure 1, Appendix A).

Ten milliliters of culture were taken and the mycelia were separated by centrifugation at 5000× *g* for 5 min. The samples were then treated in parallel with each of the protocols mentioned; three washes were performed using PBS with 30% sucrose (PanReac AppliChem, Barcelona, Spain) at pH 7.4 or with ammonium bicarbonate buffer (PanReac AppliChem, Spain) 25 mM, depending on the protocol used. The pellets were then treated with 10 µg of trypsin (Thermo-Scientific, Waltham, MA, USA) in 1 mL of PBS buffer or 1 mL of ammonium bicarbonate buffer with DTT 5 mM (Dithiothreitol; Sigma-Aldrich, St. Louis, MO, USA) and the samples were incubated for 5 min at 37 °C. In addition, images of the mycelium before and after enzymatic digestion with trypsin were recorded using a Moticam 2.0 camera coupled to the microscope (Figure 2). The samples were then centrifuged at 13,000× *g* for 10 min. The supernatants were then filtered with a 0.22 µ filter and incubated overnight at 37 °C. After the incubation period, the reaction was halted with TFA (Trifluoroacetic Acid, Thermo-Scientific, Waltham, MA, USA) at a final concentration of 0.1%. Finally, the samples were incubated in ice for 15 min. Peptides obtained after trypsin digestion were quantified using the Qubit Protein Assay Kit (Invitrogen, Waltham, MA, USA) in a Qubit^®^ 2.0 fluorometer (Invitrogen, USA) following the manufacturer’s instructions.

### 2.3. Protein Identification by LC–MS/MS

To carry out the optimized protein extraction protocol, the same procedure described above was followed using the saline phosphate buffer (PBS) with 30% sucrose (PanReac AppliChem, Spain) at pH 7.4. The biological samples used were 10 mL from the flask containing MSM plus 1% of GLU; 10 mL from the flask containing MSM plus 1% of TCW of 2 hpi (representing rapid response); and 10 mL of MSM plus 1% of TCW of 48 hpi (representing late response).

Trypsin digested samples were acidified with 100 μL 10% trifluoroacetic acid (TFA). Then, 1 mL of each acidified peptide sample was cleaned with a C18 reverse phase SEP-PAK cartridge, according to the manufacturer’s instructions. After peptide cleaning, the samples were dried, resuspended with 2% Acetonitrile (ACN) and 0.1% formic acid, and quantified using a Qubit™ Fluorometric Quantitation (Thermo Fisher Scientific). A 500 ng aliquot of each fraction was analyzed using liquid chromatography coupled to mass spectrometry (LC–MS/MS) using an Ultimate 3000 nano HPLC system (Thermo Fisher Scientific), equipped with a C-18 reverse-phase column (EASY-Spray™ PepMap RSLC C18 75 µm × 50 cm, particle size of 2 µm), coupled to an Orbitrap Exploris™ 240 mass spectrometer (Thermo Fisher Scientific, San Jose, CA, USA). Peptide fractionation was carried out at a flow rate of 250 nL/min and at 45 °C using a 120 min gradient, ranging from 2% to 95% mobile phase B (mobile phase A: 0.1% formic acid (FA); mobile phase B: 80% acetonitrile (ACN) in 0.1% FA). The loading solvent was 2% ACN in 0.1% FA and the injection volume was 5 µL.

Data acquisition was performed using a data-dependent acquisition in full scan positive mode in a range from 375 to 1200 *m*/*z*. Survey scans were acquired at a resolution of 60,000 at *m*/*z* 200, with Normalized Automatic Gain Control (AGC) target (%) of 300, a RF lens of 80%, and with an automatic maximum injection time (IT). The top 20 most intense ions from each MS1 scan were selected and fragmented via high-energy collisional dissociation (HCD). Resolution for HCD spectra was set to 15,000 at *m*/*z* 200, the normalized AGC target to 50, and the maximum ion injection time to AUTO mode. Precursors with charges of 2–5 were selected on a 2 *m*/*z* isolation window with an exclusion duration of 45 s and an HCD collision energy of 30%.

Data obtained by mass spectrometry were analyzed using Proteome Discoverer 2.4.0.305 with four different search engines (Mascot (v2.7.0), MS Amanda (v2.4.0), MSFragger (v3.1.1), and Sequest HT) against the target/decoy UniProt database of *Botrytis cinerea* (13,279 sequences; 7 July 2021) with a workflow combining processing and consensus methods. In the processing method, the precursor and fragment mass tolerance were set at 10 ppm and 0.02 Da, respectively, the maximum number of missed cleavages at 3, and acetylation in protein N-terminal, pyrrolidone from Q, deamination of NQ, and oxidation of methionine residues were considered as dynamic modifications, and carbamidomethyl (+57.021 Da) on cysteine as a static modification. Intensities were extracted from chromatographic peaks and linked to the identified peptide spectral matches (PSMs) using the Minora Feature Detector node from Proteome Discoverer.

In the consensus method, the PSMs identified using the four engines were combined and validated by calculating and setting the false positive rate (FDR) at <1% for proteins, peptides, and peptide spectral matches (PSMs). The proteins were grouped according to the identified peptide sequences Protein Grouping node. A fold change value and a Student’s *t*-test for the proteins that were in the three biological replicates in the three conditions were calculated to pinpoint differentially abundant peptides (*p*-value < 0.01). Only those proteins with ratio (R) of <0.66 and >1.5 (*p*-value < 0.01), of differences in the three conditions were considered as true differentially abundant proteins, and retained for further analyses.

For the presence/absence analysis, a protein was considered exclusive to one phenotypical condition if it was present in the three biological replicates, and was not detected in any replicate of the other conditions. No statistical analysis was performed for proteins that exhibited an absence/presence pattern between conditions. Specifically, proteins present in all the replicates of a specific assay and in none of the replicates of the rest of the conditions were considered exclusive to one condition. That means that one specific protein present in three replicates of a specific assayed condition was discounted as exclusive if it appeared only once in one of the other six replicates. Similarly, non-regulated proteins common to two conditions are those proteins present in all the replicates of these two specific assays, and in none of the three replicates of the other assay, and they are not overexpressed. On the other hand, proteins presenting measurable abundances in all the replicates of all the conditions are used to analyze differentially abundant proteins (non-regulated and overexpressed proteins) (Appendix A).

### 2.4. “In Silico” Analysis of Proteins Identified

The list of proteins identified and the datasets generated from these studies are available in the PRIDE repository, (https://www.ebi.ac.uk/pride/archive/; accessed on 1 October 2021) with the dataset identifier PXD028958. To increase the robustness of our analysis, only those proteins with a high combined protein FDR confidence level (q-value ≤ 1%) were used. Proteins present in all the replicates of a specific assay and in none of the replicates of the rest of the conditions were named as exclusive. On the other hand, proteins presenting measurable abundances in all the replicates of all the conditions were used to analyze differentially abundant proteins (non-regulated or overexpressed proteins). Using this restrictive condition, qualitative and quantitative analyses were performed. Non-regulated, exclusive, or overexpressed proteins identified under GLU and TCW conditions (representing rapid and late response) are listed in Appendix A.

Gene Ontology (GO) was annotated using the following procedure: proteins identified under GLU and TCW late response conditions (common and exclusive or overexpressed proteins), without taking into account the TCW early response condition, were annotated on the GO annotations page (“View GO Annotations”) of QuickGO (https://www.ebi.ac.uk/QuickGO/annotations; accessed on 1 October 2021), using a gene product names filter (UniProtKB accessions). Proteins without annotation in the QuickGO annotation page were annotated by GOanna and GORetriever tools from the AgBase web resource (https://agbase.arizona.edu/; accessed on 1 October 2021). Finally, GOSlimViewer from AgBase was used to provide a high-level summary of functions for GO annotated proteins [21].

In addition, the STRING protein interaction database (v11.5) (https://string-db.org/; accessed on 1 October 2021) [22] was used to generate a protein interaction network of all 2399 proteins identified (1730 proteins under GLU) and (669 proteins under TCW late response) from all the *B. cinerea* proteomics studies carried out by the research group using the same culture conditions (exclusive or overexpressed proteins (Confidence_0.4). The protein–protein network obtained was then imported into Cytoscape [23] (v3.8.2) and the clustering algorithm MCODE (v1.5.1) [24] was run to identify potential functional clusters (degree cutoff = 2; haircut; node score cutoff = 0.2; K-core = 2; max. depth = 100).

To study protein membrane associations, transmembrane domains (TMHMM v. 2.0, [25]), secretion signal (SignalP-5.0 Server, [26]), membrane associations (OutCyte 1.0 server, [27]), non-classical secretory pathways (SecretomeP 2.0 Server, [25]), GPI associated proteins (PredGPI, [28]), and protein lipidation (GPS-Lipid v1.0, [28]) were used.

## 3. Results

### 3.1. Optimization of “Shaving” Process in B. cinerea

The process to obtain surface-associated proteins, named “shaving”, is based on the action of protease that, during a short time exposure, cuts off the outer domains of the protein without affecting cell integrity. In our optimization step, two protocols were compared—one based on PBS, the other based on ammonium bicarbonate. During the optimization of the membrane surface protein extraction protocol in *B. cinerea*, different images of the mycelium were taken to ensure that the enzymatic action of trypsin was not so strong (in concentration or time) that cell integrity disintegrated; thus, preventing proteins from the whole of the mycelium cell, as well as those from the surface, being obtained (Figure 2). After the first 5 min of incubation, the hyphae showed a slightly more whitish, thin and elongated appearance in both cases; this is a product of the enzymatic action of trypsin. However, cellular integrity was maintained in both cases, showing that both buffers used, and the protocol conditions, maintain cell integrity. After extraction procedures, the peptides obtained were quantified and the yield calculated and compared. Using the PBS-based protocol, the average yield was determined at 2.02 ± 0.088, whereas the ammonium bicarbonate-based protocol yielded 1.21 ± 0.030 (Appendix A). For this reason, the PBS plus 30% of sucrose method [20] was used for this analysis of the *B. cinerea* surfactome.

### 3.2. In Silico Analysis of Identified Proteins

Three protein extracts per assayed condition (GLU 48 hpi, TCW 48 hpi and TCW 2 hpi) were identified using LC–MS/MS. From these extracts, a total of 1168 proteins were identified (Appendix A), where 1010 proteins show a high FDR confidence level (q-value ≤ 1%).

Using these filters on the 1010 proteins identified: 3 proteins were classified as exclusive or overexpressed in the GLU condition; 1 protein was exclusive or overexpressed in TCW late response condition; 6 were classified as exclusive or overexpressed in the TCW rapid response condition; 2 were identified as non-regulated proteins common to the GLU and TCW late response conditions; 2 were identified as non-regulated proteins common to the GLU and TCW rapid response conditions; and 166 proteins were classified as non-regulated proteins common to the GLU and both TCW conditions (rapid and late response) (Figure 3).

In previous studies, the percentage of the total number of proteins predicted in the genome of *B. cinerea* represented by the proteins identified in all the proteomics studies carried out, has been calculated; this proportion was found to be 10–12% [7,15]. It is also an objective of the new study reported here to determine this percentage, and by adding together the constitutive proteins detected in *B. cinerea* in all previous assays, including the surfactome proteins and taking into account the latest update of the total proteins predicted in the *B. cinerea* genome. That percentage was found to be 54%—a notably increased coverage of the genome. The genomic assembly of *B. cinerea* strain B05.10 used for this percentage calculation is the latest genome update generated by Van Kan et al. [5], which is available at the Ensembl Fungi platform (http://fungi.ensembl.org/Botrytis_cinerea/; accessed on 1 October 2021). With this update, the estimated number of genes has decreased from 16,448 to 11,707.

### 3.3. Gene Ontology Categorization

To clarify the role of the 1010 proteins identified, the identification list was categorized using Gene Ontology (Appendix A) by QuickGO and AGBASE. Proteins were categorized according to their Molecular Function (MF) and their involvement in specific Biological Processes (BP) (Figure 4). The GO categories found in 5-day-old cultures of GLU and of TCW (late response condition) were also compared.

The gene ontology classification by MF consists of 29 different categories (Figure 4A). Three categories with the highest levels of relative abundance are: (a) ion binding, associated with charged atom binding; (b) oxidoreductase activity, acting as a catalyst in a reversible redox reaction where the oxidation stage is altered; and (c) RNA binding, where proteins are attached to a RNA molecule or a portion of a molecule. No specific categories were found only in GLU or TCW assays, with minor differences between relative abundances detected in assays with each of the carbon sources.

The gene ontology classification by BP (Figure 4B) consists of 45 different categories, where the three major categories were: (a) small molecule metabolic processes, related to any biochemical pathway involving small non-encoded molecules; (b) biosynthetic processes, involving those processes where simpler substances are transformed into more complex ones; and (c) metabolic processes with cellular nitrogen compounds that involve various organic and inorganic nitrogenous compounds. In the BP categorization, the category of vacuolar transport, which includes those proteins involved in the directed movement of substances into, out of, or within a vacuole, appears only for the GLU condition, whereas categories of locomotion and cell motility, directly implied in the movement of a living cell from one place to another, were found specifically for the TCW induction.

### 3.4. Algorithm-Based Prediction of Types of Surface-Associated Proteins

The global list of identified proteins representing surface-associated proteins after the shaving process obtained 1110 hits was employed to determine the distribution of membrane associations among identified proteins (Appendix A). This in silico analysis uses different prediction algorithms to check if a protein possesses transmembrane domains (TMHMM v2.0, OutCyte 1.0 server); to check its role as a secreted protein, through classical or non-classical pathways (Signal P-5.0 Server, Secretome P 2.0 Server); and its possible relation with GPI-anchored proteins (PredGPI) or protein lipidation (GPS lipid v1.0).

First, a substrate analysis was performed whereby, if a protein presents positive results to one of the assayed algorithms, it is omitted from the next analysis. The main aim of this analysis is to establish the percentage of identified proteins without a clear relation to the membrane. From this, we determined that around 21% of the identified proteins were not directly related to surface-associated proteins, at least using those algorithms. To study the global distribution of types of association with the membrane in our results, a global analysis was performed to determine the percentage of proteins predicted by each assayed algorithm. From this analysis (Figure 5, Appendix A), the highest percentage of proteins with membrane association was predicted by the OutCyte 1.0 server, with 33%. A secretion signal (signal P) and no classical secretory pathways (secretome P) were presented by 25% of the identified proteins, whereas 54% of the proteins identified present lipid modification (palmitoylation, myristoylation, farnesylation, and geranylgeranylation. Only 1% present a positive signal for GPI-anchored proteins.

### 3.5. Cluster Analysis

To find out processes differentially activated under GLU or TCW induction, we performed a protein interaction analysis using the STRING database and Cytoscape [22,23]. Exclusive or overexpressed proteins identified under these conditions were added to the proteomics data on *B. cinerea* previously obtained by the research group under the same experimental conditions [8,12,14,15]. This set of proteins were used with the object of unravelling new protein–protein interactions that could not be identified in the previous analyses of the fungus’s secretome, phosphoproteome, membranome and phosphomembranome without its surfactome [16]. Firstly, we searched exclusive or overexpressed identified proteins against *B. cinerea* proteins in the STRING protein–protein interaction database. The interactions included direct (physical) and indirect (functional) associations, obtained from computational prediction, from knowledge of other organisms, and from interactions data in other databases. By using the identified proteins found in each condition, the analysis produced a network of 297 nodes (proteins) and 777 connecting edges (predicted associations) for TCW (Appendix A), and a network of 1337 interacting nodes and 21,966 connecting edges for GLU. These networks were obtained by means of a cluster analysis (highly interconnected regions) using the MCODE software. Clusters in a protein–protein interaction network are usually protein complexes or parts of pathways. MCODE analysis returned 16 and 29 clusters identified for the TCW late response and GLU conditions, respectively (Appendix A). Two new clusters detected under TCW condition, cluster 11 (Figure 6) and cluster 14 (Figure 7), are composed of proteins related to acetate metabolism and plant cell wall polysaccharides metabolism, respectively.

## 4. Discussion

*Botrytis cinerea* is one of the most devastating and widely studied phytopathogenic fungi. Its commercial importance is beyond dispute; and the interaction between this fungus and plant tissue has become a model in modern studies of molecular plant pathology. *B. cinerea* deploys a wide range of molecular weapons for infecting plant tissues. Among these weapons, referred to as virulence or pathogenicity factors, those whose actions are associated with signaling translation pathways play a significant role.

To obtain more knowledge of those pathogenicity factors, we initiated the analysis of the *B. cinerea* surface-associated proteins, the surfactome. This group of proteins is extremely important. Most of the experimentally-checked virulence/pathogenicity factors described in *B. cinerea* are related to those proteins [3]. Moreover, several functions that are crucial, such as nutrient transportation, receptors that trigger signaling cascades, binding factors to other cells or surfaces, and enzymatic activities, take place in the surfactome [19].

In plant pathogenic microorganisms, this subproteome is not only responsible for recognition of the host by the pathogen, but it also acts as the initial sensing system for assessing the vegetative stage of the plant, environmental conditions, and other external aspects, and it initiates the correct sequence of enzyme production that ends with fungal plant invasion. This subset of proteins has been studied previously in several bacteria but never before in a filamentous fungus. The study reported here is the first carried out on a fungal surfactome. To reach this milestone, an optimization step was added in order to check which of the assayed bacteria or yeast protocols might fit with our experimental design. With minor modifications, the best protocol was considered to be the method using PBS plus 30% sucrose [20].

Using this method, more than one thousand proteins have been identified. In this work, we have made a new calculation of the percentage of total proteins predicted for the genome represented by proteins identified in proteomics studies. Several improvements have been incorporated in this calculation; first, we include all non-redundant proteins identified in previous proteomics studies of *B. cinerea*; second, we use the latest upgrade of the *B. cinerea* genome [5] in which the total number of proteins predicted in the *B. cinerea* genome has been reduced from 16,448 to 11,707. Using these additional data, the percentage of proteins from proteomics work has been increased from 12% to 46%, and that figure increases to 54% when surfactome proteins are included.

Identified proteins were subjected to “in silico” analysis to review the different categories obtained according to each protein’s molecular function and its involvement in biological processes (Figure 4). Different prediction algorithms were used to determine the relation of these proteins to the membrane or to secretion pathways. This analysis determined that around 21% of the identified surfactome proteins seem to be cytoplasmatic proteins. In previously published proteomics studies, the proportion of cytoplasmic proteins in the surfactome of bacteria varies between 6 and 27% [29,30,31]; our results fall within that range, thus validating our finding. The explanation for this percentage is related to the methodology used, which can cause cell lysis. Other explanations proposed are the existence of different secretory pathways and/or protein secretion from membrane vesicle structures [19]. In our study, cell integrity was maintained with no trace of cell modification during their exposure to trypsin. The presence of *B. cinerea* vesicles is now under study, but it seems clear that, as happens in all “…omics” studies, the relevance of a specific protein identification must be validated in further molecular studies of site-directed mutagenesis. The development and optimization of CRISPR/Cas methodology in *B. cinerea* may reduce this bottleneck and so facilitate the characterization of the phenotype of specific genes [32].

Exclusive or overexpressed proteins identified in the *B. cinerea* surfactome in GLU and TCW conditions after 5 days were merged with proteins identified under the same experimental conditions in previous proteomic work on *B. cinerea* carried out by the research group [12,14,15,33]. Compared to previous joint proteomes analysis [16] of exclusive or overexpressed proteins identified in GLU and TCW late response conditions, the inclusion of the surfactome in the protein–protein interaction analysis with STRING produced an increased size of the predicted network in both conditions. Cluster analysis using MCODE produced 2 more clusters in the TCW condition than in previous analyses, but the number of clusters in GLU decreased from 41 to 29; however, the total number of proteins implicated was greater. The decreased number of clusters in GLU can be explained by the merger of some previously identified clusters following the increase in the number of proteins that comprise them. Lastly, two new clusters detected in the TCW condition, cluster 11 (Figure 6) and cluster 14 (Figure 7), are composed of proteins related to acetate metabolism and plant cell wall polysaccharides metabolism, respectively. It is notable that none of the exclusive or overexpressed proteins in the TCW late response condition is present in the 16 clusters identified for this condition; this is due to changes in the new STRING resource version 11.5 compared to the previous version 11. The updated version 11.5 includes an increased organism coverage to 14 094, and includes a full re-import and re-scoring of all evidence types [22].

From the identification analysis by MS/MS, three proteins were found as exclusive under GLU induction compared with TCW (Appendix A). Regulatory 26S proteasome RPN2 (A0A384J7D3; BCIN_01g10980) is an intricate, multi-subunit proteolytic mechanism involved in the selective proteolysis to control the abundance of key regulatory proteins [34]. Although the role of the 26S proteasome in *B. cinerea* remains to be elucidated, its phosphorylated form has been found in previous proteomics studies of membrane phosphorylated proteins specifically under TCW induction [15]. In our study, it has been discovered specifically in glucose conditions. Probably, this difference is related to a differential phosphorylation stage, dephosphorylated in GLU vs. phosphorylated in TCW. Since this proteasome is related to the UPR and ERAD, which are known to be essential for infection processes in several pathogenic fungi [35,36], we believe that the protease acts as an initial switch between observed GLU and TCW phenotypes.

Blast homology search has defined the SGL domain-containing protein (A0A384JRI3; BCIN_09g00770) as smp-30 gluconolactonase (EC 3.1.1.17). This enzyme has been involved in secondary catabolic pathways for glucose including the synthesis of lactones, calcium homeostasis, and other processes [37]. It hydrolyzes a variety of γ- and δ-lactones with five, six, and seven carbons. The function of this protein as a virulence factor has been suggested in previous proteomics research on *Pyrenophora teres f. teres* [38]. The presence of this protein on the surface of *B. cinerea* under glucose induction, where toxin production is at maximum, could mean that the relationship of this enzyme to pathogenicity factors is more related to the control of toxin expression than to its role as a CWDE.

The enzyme 3-phytase (A0A384J4T0; BCIN_01g02910) (EC 3.1.3.8) identified in our analysis under GLU conditions is a hydrolase that hydrolyses myo-inositol hexaphosphate (phytate), the major form of phosphate and inositol in plants that cannot be degraded by monogastric animals, as well as humans [39]. It has previously been used as an additive in foods for human and animal consumption. Phytate also acts as a chelator of mineral ions. The relevance of calcium pathways and phosphorylation in the regulation of *B. cinerea* virulence has been previously described [11,16], because the degradation of phytate may produce variations in the levels of phosphate and calcium, decreasing the concentration of this compound that is involved in plant signaling. Specifically, phytate has been related to the maintenance of basal resistance to bacterial plant pathogens [40]. These clues suggest a role for the 3-phytase identified, as a virulence/pathogenicity factor in *B. cinerea*.

From TCW cultures after 5 days (late response condition), only one uncharacterized protein was specifically identified (A0A384K1D6; BCIN_13g04850). BlastP analysis from NCBI using default settings identified this protein as alpha-L-rhamnosidase (EC 3.2.1.40), a protein that hydrolyses terminal non-reducing alpha-L-rhamnose residues. Among its substrates are terpenyl glycosides and many other natural glycosides containing terminal alpha-L-rhamnose. This enzyme has a wide range of industrial applications mainly related to the transformation of secondary metabolites in order, for example, to reduce the bitterness of citrus juices, and to enhance wine aroma by the hydrolysis of volatile terpenyl glycosides from grape skin [41]. In *Aspergillus niger*, alpha-L-rhamnosidase is located in tandem with the l-rhamnose transporter involved in the catabolism of methyl-pentose. In *B. cinerea*, alpha-l-rhamnosidase has been previously described during: “in silico” genome analysis [42]; transcriptome analysis during plant infection [43]; and in proteomics studies where the enzyme is present in the secretome of pectin and sucrose-amended media [44]. However, knockout mutants of this gene do not seem to have an effect over fungal virulence [45]. It must also be taken into account that this external domain has been found when toxin secretion of *B. cinerea* is inhibited. Those findings, plus the protein’s potential capacity to interact with complex metabolites, suggest that alpha-L-rhamnosidase plays a role as a receptor for modulating toxin secretion.

In this proteomics study, the condition of rapid response to TCW induction has been simulated by collecting surface proteins after only 2 h of induction. These surface proteins include six proteins specifically detected in this condition (referred to as TCW, 2 hpi), that could be responsible for switching from a toxin-producing stage without CWDE secretion under GLU induction, to the opposite situation, where toxin production is halted completely and CWDE are at maximum levels under TCW induction [9]. In spite of the increasing amount of molecular information on *B. cinerea*, we are still identifying some proteins that have no molecular annotation. One of these is the protein A0A384JAJ7 (BCIN 03g00010) that was identified as an uncharacterized protein. None of the common prediction algorithms revealed relevant information about its biological role in fungi. This example shows the need for more in-depth studies of the molecular weapons deployed by *B. cinerea*.

Some of these proteins identified are involved in secondary metabolism. For example, an uncharacterized protein (A0A384J6A4; BCIN_01g08220) was identified as containing a polyketide synthase (PKS) domain. This protein has been related to the production of the polyketide botcinic acid, one of the toxins of *B. cinerea* [46]. Fungal polyketides are large, multi-domain enzymes such as ketoacyl synthase (KS), acyl transferase (AT), dehydratase (DH), enoyl reductase (ER), ketoreductase (KR), and acyl carrier protein (ACP) [47]. This group contains more proteins involved in secondary metabolism. In our study, delta-aminolevulinic acid dehydratase (EC 4.2.1.24) (A0A384JDY6; BCIN_04g00200) was found in the TCW rapid response condition. This enzyme is involved in the synthesis of porphobilinogen, a common precursor of all natural tetrapyrroles, including heme, chlorophyll, and vitamin B12. Heme groups are most commonly recognized as components of hemoglobin, myoglobin, cytochromes, etc. Heme is a relevant core of cytochrome P450, involved in many complex fungal bioconversions related to toxin production in *Aspergillus parasiticus*, *A. flavus,* and *A. nidulans* [48]. In *B. cinerea*, cytochrome P450 was the first gene described in the biosynthetic pathway of botrydial, the nonspecific phytotoxin of *B. cinerea* [49], and in the synthesis of Abscisic acid [50]. The presence of both enzymes as surface proteins may indicate a role in controlling secondary metabolism, particularly toxins production. Since *B. cinerea* is affected by its own toxins that reduce its growth rate [51], we may hypothesize that there is an external localization of toxin biosynthesis, or at least that the enzymes act as a receptors in signaling cascades and/or toxin production. To support this hypothesis, there is the finding that Delta-aminolevulinic dehydratase acts to regulate the proteasome-interacting protein modulating proteolysis processes [52].

Small heat shock proteins are a widely distributed and diverse protein family. Here, one of these (A0A384J6V0; BCIN_01g09530) has been detected specifically in the TCW rapid response condition. Behind their role as stress-response proteins acting as a chaperone, proteins of this family have multiple cellular functions. In pathogenic *Candida albicans* their role as a virulence factor has been described [53]. In *Mycobacterium tuberculosis,* not only has their role as a pathogenicity factor been described, but their possible use as a therapeutic target has been proposed [54]. Another protein described under the same condition is the Multiprotein-bridging factor 1 (MBF1-A0A384JRV2; BCIN_09g00080). This protein is a transcription cofactor that forms a bridge between transcription factors and the TATA box binding protein, which is part of the basal transcription machinery, that appears to be upregulated under pathogen induction [55]. MBF1 defective mutants show reduced pathogenicity, supporting their role as virulence factors in *Magnaporthe oryzae* [56]. Our data suggest that small heat shock proteins and multiprotein-bridging factor 1 may have a role as virulence/pathogenicity factor in *B. cinerea*.

Lastly, a protein member of the spliceosome group was found under TCW 2 hpi induction: CDC40 (A0A384J4A8; BCIN_01g01530). Spliceosomes are huge, multi-megadalton ribonucleoprotein (RNP) complexes, involved in mRNA maturation and metabolism [57]. Their function is based on the process for removing introns to obtain matured mRNA. Although the splicing reaction is chemically simple, what occurs inside a cell is much more complicated; for example, the RNP maturation takes place at locations that are different from their sites of function or the recognition of splicing sites [58]. mRNA metabolism plays a crucial role in the plant versus pathogen battle, with bullets being fired from both fronts. Plant small RNAs are considered to be mandatory elements of plant defense [59]. For example, *Arabidopsis* microRNA has been shown to trigger immune responses against *B. cinerea* [60]. It has also been reported that *B. cinerea* is able to produce sRNA and downregulate transcripts of defense genes [61]; for example, *Botrytis* sRNA Bc-siR37 suppresses plant defense genes of *Arabidopsis* [62]. In this environment, the presence of CDC40 may indicate a role in the management of *Botrytis* sRNA outside the cell or in the suppression of plant sRNA counterattack.

## 5. Conclusions

The present paper reports the first documented study of the surfactome of this filamentous fungus. An optimized protocol has been developed to obtain the surface-associated proteins during two pathogenic stages and in both rapid and late response conditions, using glucose and deproteinized tomato cell walls as a sole carbon source. Both stages have been well-characterized previously [9] in terms of CWDE and toxin production. In our study, we identified 1010 proteins that are components of the surfactome of the fungus; these have been categorized by gene ontology and protein–protein interactions, and new potential pathogenicity/virulence factors in *B. cinerea* have been revealed. From these data, the percentage of total proteins predicted from the genome of the fungus represented by those proteins identified in proteomics studies is calculated to have increased from 12% to 54%. This is a significant increase in knowledge and the new data should be crucial for a better understanding of the biological activity and pathogenicity of *B. cinerea*. Because the surfactome we are describing is highly exposed to the plant tissues and environmental conditions, it presents numerous opportunities for the fungus to interact with external elements; and these interactions are likely to offer the best targets for the development of fungicides and infection-prevention treatments. Among the proteins obtained and identified in this study, there may be the one that is critical for the development of new environmentally friendly strategies to help prevent the devastating crop losses caused by *B. cinerea*.

## Figures and Tables

**Figure 1 jof-07-01021-f001:**
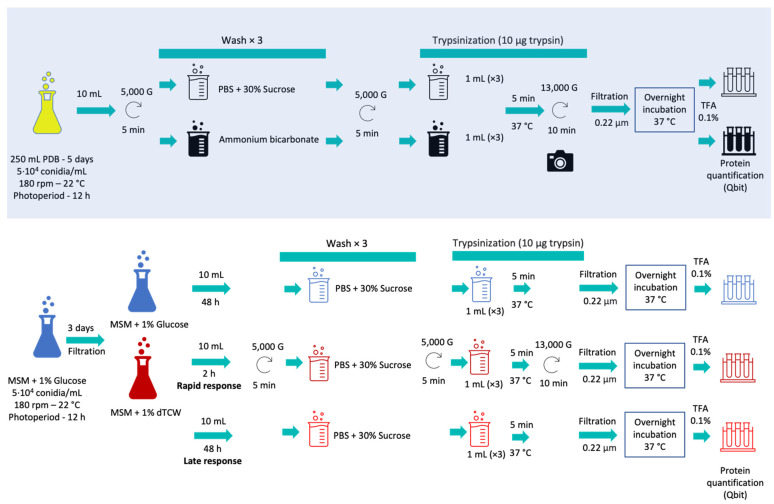
Schematic protocol followed during surfactome optimization (with blue shadow) and during the experimental work with glucose and deproteinized tomato cell wall as sole carbon sources, representing rapid and late responses.

**Figure 2 jof-07-01021-f002:**
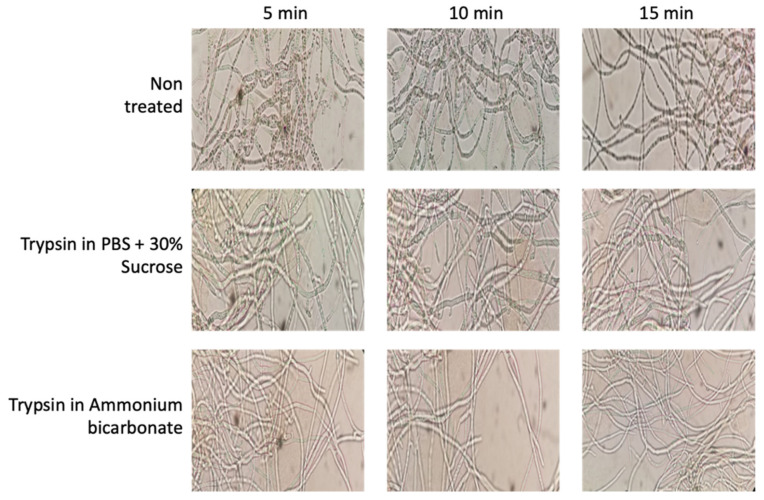
Effects of trypsin treatments on cell integrity using PBS plus sucrose and ammonium bicarbonate buffers during 5, 10, and 15 min, showing the maintenance of cell integrity during the protocol (Motic Microscope, Moticam 2.0 camera using 40× Objective).

**Figure 3 jof-07-01021-f003:**
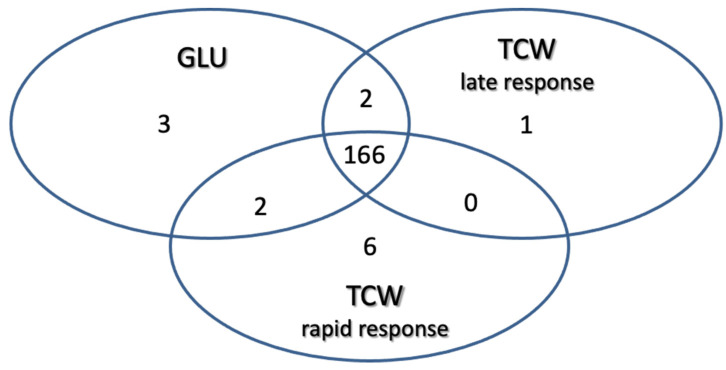
Classification of identified proteins as exclusive or overexpressed in each assayed condition and non-regulated proteins common to two or three conditions.

**Figure 4 jof-07-01021-f004:**
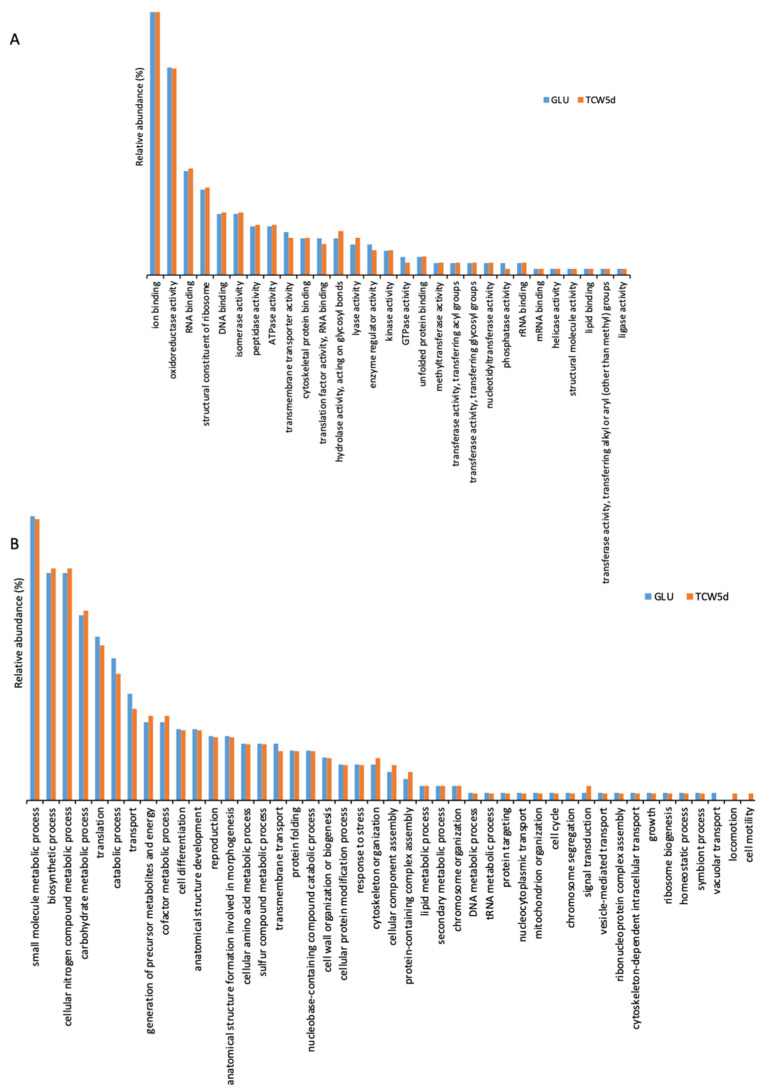
Gene ontology categorization of the identified surfactome proteins by their molecular function (**A**) and their involvement in specific biological processes (**B**).

**Figure 5 jof-07-01021-f005:**
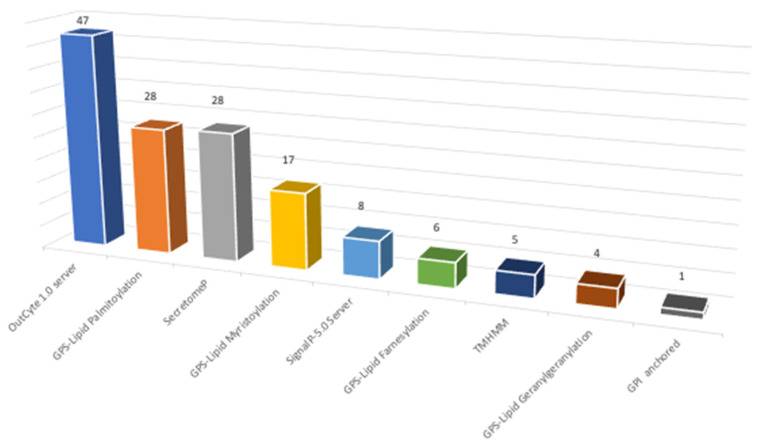
Characterization of surfactome proteins according to their type of association with membrane (data in percentage).

**Figure 6 jof-07-01021-f006:**
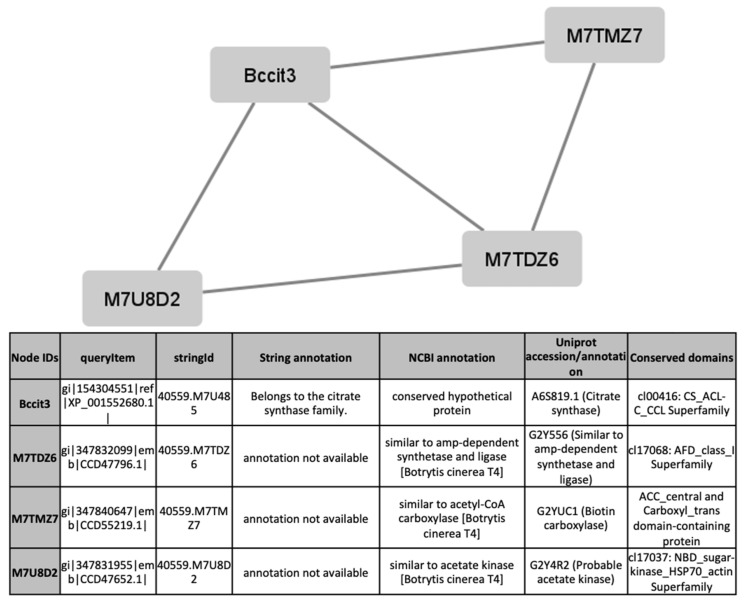
Cluster 11, found in the TCW condition using the STRING and MCODE, containing exclusive and overexpressed proteins of the secretome, membranome, phosphoproteome, phosphomembranome, and surfactome in the TCW condition, and composed of proteins related to acetate metabolism.

**Figure 7 jof-07-01021-f007:**
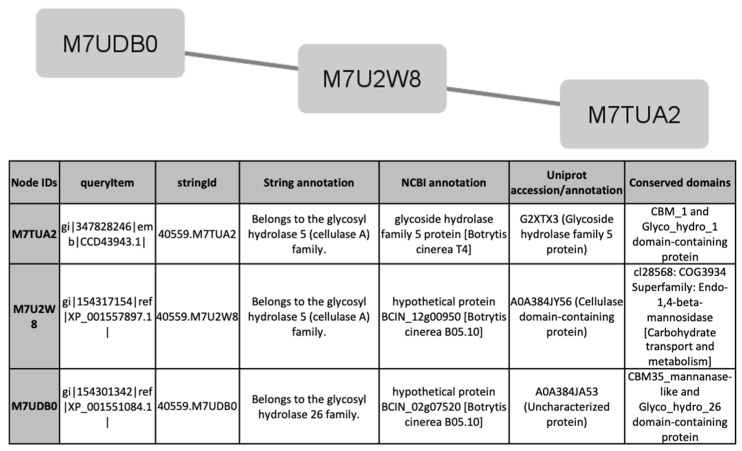
Cluster 14, found in the TCW condition using the STRING and MCODE, containing exclusive and overexpressed proteins of the secretome, membranome, phosphoproteome, phosphomembranome, and surfactome in the TCW condition, and composed of proteins related to metabolism of plant cell wall polysaccharides.

## Data Availability

Mass spectrometry proteomics data were deposited to the ProteomeXchange Consortium via the PRIDE partner repository, with the dataset identifier PXD028958 and 10.6019/PXD028958.

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
