# Peer review of "Unravelling the Initial Triggers of *Botrytis cinerea* Infection: First Description of Its Surfactome"

_jof, 2021, doi:10.3390/jof7121021_

Round 1
Reviewer 1 Report
The MS entitled "Unravelling the initial triggers of Botrytis cinerea infections.
Botrytis cinerea surfactome" by Escobar-Niño et al., characterized the surfactome of B. cinerea. The MS found some interesting results showing some specific proteins associated with pathogenesis of B. cinerea. These data may be helpful for further protein analysis of B. cinerea. However, the writing of the MS is redundant and rough, and some issues in the MS are required to addressed. Generally, the MS is required to be further polished.
some concerns:
- The discussion is too long. It is hard to get the major points for the authors, try to summarize the major finding and point out their importance.
- Please include the “Gene ID” (https://fungi.ensembl.org/Botrytis_cinerea/Info/Index) for each surface protein in the MS. This information may be very helpful for the readers
- L-45 delete “a”
- L-55 should be “factors”
- L-30 change “mycovirus-free” to “RnHV2-free”, there is no evidence that Rn400 contains no other viruses.
- L-65 effectors generally means some specific proteins associated with plant-microbe interactions, not receptors etc.
- L-72 “B. cinerea” should be italicized, please check other places in the MS.
- L-93 should be “interests”
- L-104-105 should be “fungus”
- L-131-132 “Streptococcus pneumonia, Candida parapsilosis and Candida tropicali” should be italicized.
- L-252-253 please reword the two sentences.
- L-264-265 reword the sentence.
Author Response
Reviewer 1 comments:
Botrytis cinerea surfactome by Escobar-Niño et al., characterized the surfactome of B. cinerea. The MS found some interesting results showing some specific proteins associated with pathogenesis of B. cinerea. These data may be helpful for further protein analysis of B. cinerea. However, the writing of the MS is redundant and rough, and some issues in the MS are required to addressed. Generally, the MS is required to be further polished.
- The discussion is too long. It is hard to get the major points for the authors, try to summarize the major finding and point out their importance.
Following the reviewer’s suggestion, the discussion has been reduced. However, since the other reviewer suggested to increase the information of each identified protein the authors had tried to maintain a balance between both criteria.
- Please include the “Gene ID” (https://fungi.ensembl.org/Botrytis_cinerea/Info/Index) for each surface protein in the MS. This information may be very helpful for the readers
Following the reviewer’s instruction “Gene ID” has been included in the MS
- L-45 delete “a”
Following the reviewer’s instruction “a” has been deleted
- L-55 should be “factors”
Following the reviewer’s instruction “factors” has been included
- L-30 change “mycovirus-free” to “RnHV2-free”, there is no evidence that Rn400 contains no other viruses.
We do not understand this suggestion, the term virus does not appear in our paper.
- L-65 effectors generally means some specific proteins associated with plant-microbe interactions, not receptors etc.
Following the reviewer’s instruction and to avoid misunderstandings, the term “efectors” has been replaced by “components”
- L-72 “B. cinerea” should be italicized, please check other places in the MS.
Following the reviewer’s instruction “B. cinerea” has been italicized, the text has been revised accordingly.
- L-93 should be “interests”
Following the reviewer’s instruction “interests” has been included
- L-104-105 should be “fungus”
Following the reviewer’s instruction, “fungi” has been replaced by “fungus”
- L-131-132 “Streptococcus pneumonia, Candida parapsilosis and Candida tropicali” should be italicized.
Following the reviewer’s instruction “Streptococcus pneumonia, Candida parapsilosis and Candida tropicali” have been italicized.
- L-252-253 please reword the two sentences.
Following the reviewer’s instruction, the sentences have been modified
- L-264-265 reword the sentence.
Following the reviewer’s instruction, the sentence has been modified
Reviewer 2 Report
In this paper, authors studied the ‘surfactome’ of B.cinerea under the condition of host (mimic) induction and aimed to reveal new potential pathogenic factors by proteomic research methods. This study is well designed, scientifically analyzed and obtained some results. However, some problems in the article need to be revised or explained.
- The quality of the English is insufficient, such as some sentences are too long to understand, grammatical errors, etc.. Please have a native English speaker proofread the manuscript or use an online editing service.
- Does ‘surfactome’ not include ‘secretome’? Secreted proteins have been considered to play important roles in pathogen-host surface interaction.
- The amount of proteins extracted under different condition should be verified by experiments, such as SDS-PAGE.
- Different buffers seem to have no effect on the integrity of mycelial cells. Were the different protein concentrations obtained due to different reaction conditions of trypsin? Have you tried other reaction conditions? For example, try the most suitable reaction conditions for trypsin.
- Only trying to change one condition (buffer) seems too simple for the optimization of the method. Do other conditions affect the extraction of protein? For example, washing times, trypsin dosage, reaction time, etc.
- Does trypsin treatment time prolonged to 10 min or 15 min affect cell integrity? Why choose the enzyme treatment time of 5min?
- In the result part, too much content is the description of experimental methods or analytical methods. Instead, some of the results are placed in the method section, such as Figure 2.
- In Figure 3, only total of 180 proteins was shown. Why did the author express 1010 identified proteins for analysis in this article?
- I'm surprised why only a small amount of protein differed between different treatments?
- Why do you choose pie chart to represent the predictive analysis of identified proteins by different analysis methods? The pie chart is more suitable to represent the proportion of each part (the total is 1 or 100%).
- figure 6 and figure 7 were not mentioned in the result section.
- Authors should analyze the identified differential proteins (only a few) in detail, such as protein domain or functional domain, role in pathogenesis, etc, and show the analysis results in the paper.
Other minor revisions:
Line 14, Dose ‘virulence stage’ mean ‘infection stage’?
Line 15, what dose ‘the first approach to filamentous fungi surfactome’ mean? The specific function of this approach should be indicated.
Line 18-19, is this radio correct?
Line 22, do you mean to use these proteins as a targets to develop new strategies?
Line 32, what dose ‘most relevant’ mean? strategies
Line 36, change to B.cinerea.
Line 41, 1.3%?
Line 114, TCw medium is very important for understanding the purpose of this article, so the preparation method should be briefly described in the method section.
Line 131-132, Latin should be written in italics.
Line 268, ‘por’?
Line 296, How did you get the ratio of 54%?
Lin 290-299, This part is more suitable for discussion.
Figure 4, No ordinate value.
Author Response
Reviewer 2:
- The quality of the English is insufficient, such as some sentences are too long to understand, grammatical errors, etc. Please have a native English speaker proofread the manuscript or use an online editing service.
Following the reviewer’s suggestion, the present version of the MS has been corrected by a professional editing service. Several modification has been included in the new version of the MS.
- Does ‘surfactome’ not include ‘secretome’? Secreted proteins have been considered to play important roles in pathogen-host surface interaction.
Of course, secreted proteins play a crucial role in fungal biology and especially during the infection cycle of phytopathogenic fungi. This was highlighted by the corresponding author during the first determination of the 2-DE profile of the B. cinerea secretome (https://doi.org/10.1002/pmic.200900408) using the same experimental approach. Proteins from the secretome, membranome and surfactome have a relationship with membranes, at least during a short time-frame, because the association with membranes is one stage of the secretion process. In our approach, mycelia were harvested by filtration and washed several times to remove the presence of secretome proteins.
- The amount of proteins extracted under different condition should be verified by experiments, such as SDS-PAGE.
The amount of proteins extracted under different conditions has been verified by two different procedures. Firstly, extracted proteins were quantified using the specific fluorometer protocol (Qubit, thermo Fisher Scientific). Results are listed in Supplementary Material Table S1. Secondly, the peptide content was measured using the MS/MS Platform, “A 500 ng aliquot of each fraction was analyzed using liquid chromatography coupled to mass spectrometry (LC-MS/MS) using an Ultimate 3000 nano HPLC system (Thermo Fisher Scientific), equipped with a C-18 reverse phase column (EASY-Spray™ PepMap RSLC C18 75µm x 50 cm, 2 µm of particle size), coupled to an Orbitrap Exploris™ 240 mass spectrometer (Thermo Fisher Scientific, San Jose, CA)”; L 176-181. As a result of peptide digestion, the particles obtained are too small to be resolved in a common SDS-PAGE procedure. Hence this is considered an unsuitable method for validating peptide content. HPLC plus MS/MS is preferred.
- Different buffers seem to have no effect on the integrity of mycelial cells. Were the different protein concentrations obtained due to different reaction conditions of trypsin? Have you tried other reaction conditions? For example, try the most suitable reaction conditions for trypsin.
- Only trying to change one condition (buffer) seems too simple for the optimization of the method. Do other conditions affect the extraction of protein? For example, washing times, trypsin dosage, reaction time, etc.
- Does trypsin treatment time prolonged to 10 min or 15 min affect cell integrity? Why choose the enzyme treatment time of 5min?
We thank the reviewer for these suggestions about the optimization process. I would like to summarize my answers to the questions proposed. During enzymatic reaction there are multiple variables that may affect the results, e.g. amount of biomass used, temperature, buffer, enzyme concentration, etc. The number of variables to adapt may be increased almost infinitely. In our work, the purification, study and analysis of the surfactome, those variations have been widely discussed by Olaya-Abril et al in 2014 (https://doi.org/10.1016/j.jprot.2013.03.035), including the most suitable reaction conditions for trypsin, washing times, trypsin dosage, reaction time, redigestion, trypsin treatment time etc. For optimization of the characterization of the B. cinerea Surfactome, we apply a mixture of the common protocols from the bibliography, mainly those used with Streptococcus pneumonia and Candida spp. (L 133-134). So as not to prolong the optimization process, we decide to adopt the common features and vary only the big difference, the buffer used. The duration of the treatment was 5 min, but we checked that 10 min or 15 min treatment does not affect cell integrity. The selection of 5 min was based on i) previous references, ii) preventing the digestion of cell wall proteins that may be digested if the time is prolonged, and iii) the results obtained measured with Qbit at 5 min was enough for MS/MS characterization.
- In the Results part, too much content is on analytical methods. Instead, some of the results are placed in the Method section, such as Figure 2.
Following the reviewer’s suggestion, results have been moved from the Method section to the Results section (figure 2). In addition, the description of experimental methods that appears in the Results section has been deleted or moved to the Methods section. However, it should be noted that improvement of the methodology is among the objectives of our work; therefore a lot of content is devoted to methods.
- In Figure 3, only total of 180 proteins was shown. Why did the author express 1010 identified proteins for analysis in this article?
We agree with the reviewer that this is noteworthy. However, this question is explained in the Results section (lines 293-314). This is usual in proteomics studies where the global list of proteins identified is reduced when statistical analyses are applied. In our case, from the 1168 proteins identified, 1010 was the total number of identified proteins with a High combined protein FDR confidence level (q-value ≤1%), which were those considered as statistically relevant for the identification. From these identified proteins, we were only interested in those presenting differences between conditions (exclusive or over expressed proteins). To analyze exclusive and overexpressed proteins, we applied additional filters to the 1010 identified proteins that reduced the number of proteins used.
Exclusive proteins of one condition were those proteins present in the three replicates of a specific assay and in none of the replicates of the rest of the conditions. This filter used for exclusive proteins excluded those proteins not present in all the replicates of the condition of interest, or those proteins present only once in one of the other six replicates of the other two conditions. Similarly, to analyze overexpressed proteins we had to determine which were the proteins common to all the three conditions assayed. So, proteins common to two conditions were those proteins present in all the replicates of these two specific assays and in none of the three replicates of the other one. In addition, proteins common to all conditions were those proteins present in all the replicates of the three conditions assayed, excluding those identified in only one or two of the replicates.
Finally, proteins presenting measurable abundances in all the replicates of all the conditions were used to analyze differentially abundant proteins, differentiating common proteins as overexpressed or non-regulated proteins. This restrictive analysis may decrease the number of relevant proteins in our assay but it ensures the relevance of each identification.
- I'm surprised why only a small amount of protein differed between different treatments?
We agree with the reviewer that this could seem surprising, but it is a usual result in proteomic studies, especially when analyzing subproteomes expressed when different carbon sources are used. The percentage of differentially detected proteins in a previous reported analysis of the phosphoproteins identified using Glu and TCW (only late response) conditions and published by the corresponding author was about 0.82% and 12.85% (10.1016/j.jprot.2016.03.019). In the present manuscript we are comparing three conditions (Glu, TCW rapid response and TCW late response) instead of the previously compared two conditions (Glu and TCW late response). In addition, the surfactome is a subproteome of another subproteome, the membranome, where we analyze only a small number of the membrane proteins. All of these reasons contribute to diminish the percentage of differentially detected proteins to 0.1%- 0.59%.
- Why do you choose pie chart to represent the predictive analysis of identified proteins by different analysis methods? The pie chart is more suitable to represent the proportion of each part (the total is 1 or 100%).
Following the reviewer’s suggestion, the pie chart of figure 5 has been replaced by a bar graph.
- figure 6 and figure 7 were not mentioned in the result section.
The Results section shows the main results obtained from the cluster analysis. The results obtained were 16 and 29 different clusters identified in the TCW late response and GLU conditions respectively. In a deeper approach, the clusters 11 and 14 obtained under TCW were analyzed in the Discussion section. However, and following the reviewer’s suggestion, both figures have been moved to the Results section.
- Authors should analyze the identified differential proteins (only a few) in detail, such as protein domain or functional domain, role in pathogenesis, etc., and show the analysis results in the paper.
We agree with the reviewer’s suggestions. In fact, this kind of analysis is exactly what we do in the Discussion section. Specific and/or overexpressed proteins in each of the assayed conditions are described in the Discussion section, giving its accession number, the “gene ID” in the B. cinerea genome database.
In those cases where the proteins were identified as “hypothetical” or “predicted” proteins, blastp results are reported to determine the putative function of non-characterized or non-annotated proteins. Finally, their potential role in the pathogenicity cycle is hypothesized based on previous bibliographical evidence.
Other minor revisions:
- Line 14, Dose ‘virulence stage’ mean ‘infection stage’?
Following reviewer instruction and to avoid misunderstandings, the term “virulence” has been replaced by “infection”
- Line 15, what dose ‘the first approach to filamentous fungi surfactome’ mean? The specific function of this approach should be indicated.
The sentence “the first approach to filamentous fungi surfactome” means exactly that the surfactome has been characterized previously in 10 gram positive bacteria, 8 gram negative bacteria, and 6 eukaryotes (L 102-103) however none of them are filamentous fungi. Thus, this is the first proteomic approach that describes and analyze the surfactome obtained from a filamentous fungi. the specific function of this approach is indicated in lines 9-14.
- Line 18-19, is this radio correct?
Yes, the ratio is correct. This percentage has been calculated using the last genome update generated by Van Kan et al. (https://doi.org/10.1111/mpp.12384), which is available at the En-sembl-Fungi platform (http://fungi.ensembl.org/Botrytis_cinerea/). This calculation is deeply explained in results (L290-299) and discussion (L396-410)
- Line 22, do you mean to use these proteins as a targets to develop new strategies?
Following the reviewer´s suggestion the sentence has been modified.
- Line 32, what dose ‘most relevant’ mean? strategies
It means that B. cinerea has been described as the second relevant fungal phytopathogen ("The Top 10 fungal pathogens in molecular plant pathology”, DOI: 10.1111/j.1364-3703.2011.00783.x) only behind M. orizae that is mainly a rice pathogen.
- Line 36, change to B.cinerea.
Following reviewer instruction, Botrytis cinerea has been change to B. cinerea.
- Line 41, 1.3%?
This typographical error has been corrected
- Line 114, TCW medium is very important for understanding the purpose of this article, so the preparation method should be briefly described in the method section.
Following the reviewer’s suggestion, the method has been included.
- Line 131-132, Latin should be written in italics.
This typographical error has been corrected
- Line 268, ‘por’?
Following reviewer instruction, ‘por’ has been change to ‘per’ (line 280)
- Line 296, How did you get the ratio of 54%?
Dividing all the identified proteins in all the proteomics approaches to B. cinerea between the predicted proteins in the genome update generated by Van Kan et al. (https://doi.org/10.1111/mpp.12384) and multiplying it by 100. For this calculation we have removed duplicated proteins of the identified ones in different papers. This information is explained in results and discussion sections.
- Line 290-299, This part is more suitable for discussion.
Following reviewer instruction’s, this part has been included in the Discussion section . A brief mentionwith the obtained data has been maintained in the result section.
- Figure 4, No ordinate value.
The values are ordinated in terms of relative abundance (%) according to glucose condition. Datailed data are referenced to Supplementary Material Table S3.
Round 2
Reviewer 2 Report
The revised manuscript has addressed all of my concerns.